# Synthesis and Characterization of a Two-Station Two-Gate Calix[6]arene-Based [2]Catenane

**DOI:** 10.3390/molecules30030732

**Published:** 2025-02-06

**Authors:** Margherita Bazzoni, Francesco Rispoli, Sara Venturelli, Gianpiero Cera, Andrea Secchi

**Affiliations:** Dipartimento di Scienze Chimiche, della Vita e della Sostenibilità Ambientale, Università di Parma, Parco Area delle Scienze 17/A, I-43124 Parma, Italy; margherita.bazzoni@univ-nantes.fr (M.B.); francesco.rispoli1@unipr.it (F.R.); gianpiero.cera@unipr.it (G.C.)

**Keywords:** calix[6]arenes, [2]catenanes, [2]rotaxanes, viologen salts, molecular machines, azobenzene, stilbene

## Abstract

The design, construction, and operation of devices and machines at the molecular scale using the bottom-up approach captivates a lot of interest in nanoscience. Particularly intriguing are interlocked molecular architectures, which are ideal candidates for these aims. [n]Pseudorotaxanes, [n]rotaxanes, and [n]catenanes serve as versatile prototypes for constructing molecular machines because they can be engineered to execute a diverse range of functions, including mechanical-like movements in response to chemical, photochemical, or electrochemical stimuli. The study explores the synthesis and characterization of a two-station two-gate calix[6]arene-based [2]catenane. Building on prior work with calix[6]arene-based Mechanically Interlocked Molecules (MIMs), this research integrates two functional gates—an azobenzene unit and a stilbene unit —into a two-station “track” ring. The synthesis employed threading and capping strategies to prepare the precursor [2]rotaxane isomers 12(*azo-up*) and 12(*azo-down*). Challenges in the deprotection of TBS groups led to the adoption of a supramolecular-assisted approach for the direct synthesis of the desired pseudorotaxane. The final catenation reaction, using a trans-stilbene-based bisacyl chloride as the “clipping unit”, afforded the [2]catenane C3(*azo-down*) in 25% yield after purification. Mass spectrometry and NMR spectroscopy confirmed the successful synthesis and orientation of C3(*azo-down*).

## 1. Introduction

Mechanically Interlocked Molecules (MIMs), such as [n]rotaxanes, [n]catenanes, and knots, are molecular architectures where components are linked not through chemical bonds but by their mechanical interlocking [1,2,3,4,5]. The number of components in such species is denoted by square brackets before their name. A distinctive characteristic of MIMs is their ability to exhibit controlled mechanical motion between components when triggered by external energy inputs. This property enables MIMs to function as prototypes for molecular machines, with their operations governed by the chemical information encoded within their structure [6,7,8,9,10].

Catenanes, characterized by two or more interwoven macrocycles that cannot be separated without breaking covalent bonds [11], have garnered significant interest in recent years [12,13,14]. A key feature of these compounds is the potential for relative motion between their constituent rings [15,16,17], a property that is pivotal to developing rotary molecular motors [18,19,20]. In [2]catenanes-based rotary motors, unidirectional motion is usually achieved through successive orthogonal chemical transformations applied on one or both rings. For example, Leigh and coworkers demonstrated catenane systems in which the smaller ring moves stepwise between different binding sites on a larger interlocked ring, driven by light, heat, or chemical stimuli that change the relative affinities of the components [15,21].

Despite their potential, synthesizing catenanes remains challenging due to the inherent difficulty of spatially arranging independent molecular components to form the desired mechanical bonds reliably. Traditionally, most reported MIMs, including catenanes, have been synthesized using *passive template approaches*, which exploit non-covalent interactions to preorganize components into a [n]pseudorotaxane precursor. This preorganization facilitates the subsequent formation of mechanical bonds [22,23]. More recently, the *active template method*, pioneered by Leigh and co-workers [24], has emerged as a complementary and innovative strategy for the synthesis of these MIMs [25]. This approach involves incorporating a reactive unit—typically a metal ion—into one of the catenane components, enabling it to simultaneously orchestrate the spatial arrangement of the components and catalyze the formation of the mechanical bond.

Calix[n]arenes are versatile platforms in host-guest chemistry, known for their ability to bind charged and neutral species selectively [26]. Over the past decade, calix[6]arene derivatives have been employed as a scaffold for constructing MIMs. For instance, it has been shown that electron-deficient axially symmetric guests with suitable chemical features can thread through the calix[6]arene annulus to form supramolecular assemblies, including pseudorotaxanes, rotaxanes, and catenanes [27]. Our research group pioneered the use of a heteroditopic calix[6]arene, functionalized on its larger rim with three *N*-phenylureido units (**TPU**, Figure 1a), as a host for a wide range of *N*,*N*′-dialkylviologen salts in low-polarity solvents, forming [2]pseudorotaxane complexes, whose stability is affected by solvent polarity [28]. The redox-active viologen core provides additional control, as single-electron reduction induces rapid disassembly of the complex [29]. Building on these calix[6]arene-based [2]pseudorotaxanes, several symmetrical and unsymmetrical [2]rotaxanes were successfully synthesized [28,30,31]. Furthermore, by increasing the number of binding sites on the axial component, these syntheses were extended to more complex [3](pseudo)rotaxane architectures [32,33].

The first example of a calix[6]arene-based [2]catenane was reported by Neri et al. in 2013 [34]. Later, our research group synthesized the [2]catenane C1 (Figure 1b) using a ring-closing metathesis reaction to link the unsaturated ends of the axle component in the corresponding [2]pseudorotaxane complex [35]. While **C1** validated the feasibility of our synthetic strategy for catenane formation, it could not enable component shuttling under electrochemical stimuli, as it featured only a single binding site for the calix[6]arene macrocycle. This limitation was subsequently addressed in **C2** (Figure 1c), which incorporated a “track” ring containing two distinct binding sites —bipyridinium and ammonium [36]. Building on this progress, we report the synthesis of **C3**(*azo-down*), a two-station two-gate calix[6]arene-based [2]catenane (Figure 1d). The “track” ring of this catenane integrates an azobenzene and a stilbene unit at the lower and upper rim of the macrocycle, which separates the bipyridinium (V-station) and ammonium (N-station) binding sites. In perspective, **C3**(*azo-down*) serves as a prototype rotary motor, where the two isomerizable and orthogonal “gate” units control the directionality of the calix[6]arene macrocycle along the track ring in response to proper external stimuli.

## 2. Results

### 2.1. Synthesis of Axle **10**

The two-station track ring in catenanes **C2** (Figure 1c) cannot adequately discriminate if the shuttling motion of calix[6]arene **TPU** occurs preferentially in a clockwise or counterclockwise direction. The tools typically used to trigger and evaluate this movement, such as EPR and cyclic voltammetry, can only discriminate between the starting and the arrival “stations” (binding sites) but cannot determine the directionality of the components’ motion [36]. To gain such crucial information, the track ring must integrate “gates” or functional groups capable of changing their molecular geometry in response to specific stimuli. According to the geometry adopted, a gate can or cannot allow the shuttling of **TPU** between the two stations. The alternating placement of gates and stations within the track ring could, in principle, enable the controlled directional movement of **TPU**. Specifically, based on each gate’s non-hindrance (*NH*) or hindrance (*H*) status, the calixarene macrocycle may only move clockwise or counterclockwise, as depicted in Figure 1d. When a gate is in the *NH* state, the shuttling kinetic will be exclusively determined by the wheel’s directional preference along the track. Another key factor to be considered is that the length of the track ring must be appropriate to allow **TPU** to move freely along it. Furthermore, if the starting axle component is too short, its cyclization will be disfavored due to the excessive torsion needed for the two chains to react. Conversely, an axle that is too long leads to excessive chain flexibility, decreasing the probability of an intramolecular ring-closing reaction.

A potential axle component meeting the above criteria should incorporate, within its “track”, a redox-active viologen unit (V-station) and an ionizable amino group (N-station), separated by a “gate” unit (G1) via alkyl spacers of adjustable length, as shown in Figure 2. Additionally, the axle should feature terminal functional groups that facilitate a straightforward cyclization reaction with a second difunctional gate unit (G2). However, it is important to note that the asymmetry of molecular components like **TPU** further complicates the synthesis of such Mechanically Interlocked Molecules. With this non-palindrome host, the catenation reaction can yield two distinct orientational catenane isomers when an asymmetric axle like the one designed is used as a precursor. The orientational isomers differ in the relative position of the gates with respect to the macrocycle rims, as illustrated in Figure 2.

Based on these insights, we aimed to synthesize axle **10** (Figure 1), which incorporates a photoisomerizable azobenzene “gate” (G1) positioned between the V- and N-stations and features hydroxyl groups at both termini. The azobenzene unit was chosen for its well-established *Z*/*E* photoisomerization properties [37,38], while the hydroxyl groups facilitate rapid and efficient esterification reactions with various dicarboxylic species. All components of the axle are separated by C6 alkyl spacers, offering synthetic versatility.

**Scheme 1 molecules-30-00732-sch001:**
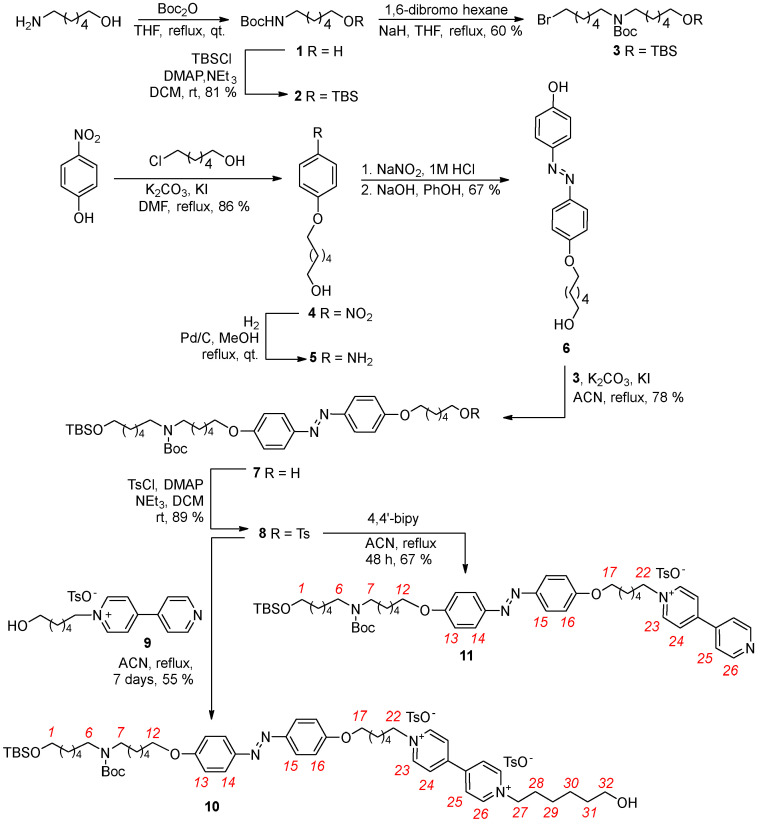
Synthesis of axle **10** and semi−axle **11**. Red numbers indicate axle proton labeling.The second intermediate for the preparation of **10**, the azobenzene precursor **8**, was prepared via a five-step procedure starting from commercially available *p*-nitrophenol (Figure 1). In the first step, the nitro derivative **4** was synthesized in an 86% yield by reacting *p*-nitrophenol with 6-chlorohexan-1-ol in refluxing dry dimethylformamide (DMF) for 24 h, using K_2_CO_3_ as the base and KI as the catalyst [39]. After a reaction workup, **4** is obtained as a pure yellow solid in 86% of yield. The nitro group of **4** was then quantitatively reduced to obtain **5** using hydrazine monohydrate as the hydrogen source and Pd/C as the catalyst in refluxing methanol for 24 h. The subsequent diazotization of **5** with phenol produced the azobenzene intermediate **6** in 70% yield. After NMR identification, **6** was regioselectively alkylated on its phenolic group with **3** in refluxing dry acetonitrile, using K_2_CO_3_ as the base and KI as the catalyst. The desired azobenzene derivative **7** was isolated as a waxy yellow solid with a 78% yield after chromatographic purification. The identity of **7** was verified by NMR and MS measurements (see Appendix A). In the subsequent step, the terminal hydroxyl group of **7** was converted into a tosylate by reacting with *p*-toluenesulfonyl chloride in dry dichloromethane. After chromatographic separation, the resulting tosylate **8** was isolated as a yellow solid with 89% yield (see Appendix A). Compound **8** was then used to synthetize axle **10** and semi-axle **11**. Reaction with the known pyridylpyridinium salt **9** [40] in refluxing acetonitrile for seven days yielded axle **10** in 55%. Similarly, reaction with 4,4′-bipyridine under the same conditions for 48 h produced semi-axle **11** of 67%.

A retrosynthetic analysis identified the *N*-protected secondary amine **3** and the azobenzene derivative **8** as the key precursors. The synthesis of **3** started from the commercially available 6-aminohexan-1-ol. The *N*-Boc-protected derivative **1** was synthesized in quantitative yield by reacting the 6-amino-1-hexanol with di-*tert*-butyl dicarbonate (Boc_2_O) in a 1:2 stoichiometric ratio. The reaction was performed by refluxing anhydrous tetrahydrofuran (THF) for 24 h, yielding a colorless oil identified as **1** by ^1^H NMR spectroscopy. In the next step, **1** was reacted with *tert*-butyldimethylsilyl chloride (TBSCl) in dry dichloromethane at room temperature for 24 h to afford the protected alcohol **2** with an 81% yield. Compound **2** was subsequently converted into the *N*-protected *O*-protected intermediate **3**. The reaction was carried out under an inert atmosphere in dry THF, using sodium hydride as the base to deprotonate the NH group of **2**. To promote the formation of the monosubstituted product, a 1:3 stoichiometric excess of 1,6-dibromohexane was used. Compound **3** was isolated as a colorless oil in 60% yield following chromatographic purification. Its identity was confirmed by ^1^H NMR and MS measurements (see experimental part and Appendix A).

The ^1^H-NMR spectra of semi-axle **11** and axle **10** are presented in the stack plot in Figure 3. In the high-field region, both spectra display four singlets corresponding to the TBS and Boc protecting groups and the methyl group of the tosylate anion, observed at 0.07, 0.90, 1.45, and 2.35 ppm, respectively. The sharp signal of the Boc group partially overlaps with the multiplets originating from the methylene groups of the alkyl spacers, which appear in the 1.2–2.2 ppm range. In the mid-field region (3 to 5 ppm), several partially overlapped triplets are observed. These resonances arise from the methylene groups adjacent to the V- and N-station and the azobenzene unit of the axles. At approximately 3.2 ppm, two overlapped triplets, integrated for a total of four protons, are assigned to the methylene protons adjacent to the Boc-protected N-station, labeled as (*6*) and (*7*) in Figure 1. The methylene group (*1*), adjacent to the TBS-protected terminal hydroxyl, resonates as a triplet centered at 3.62 ppm in **11** (see Figure 3a). In **10**, this signal is overlapped with the triplet assigned to methylene (*32*) (see Figure 3b). The methylene groups (*12*) and (*17*), positioned adjacent to the azobenzene unit, give rise to a triplet at 4.08 ppm in **11** and to a multiplet around 4 ppm in **10**. Additionally, the methylene group adjacent to the V-station resonates in **11** as a single triplet at 4.69 ppm (*22*), integrating two protons. In **10**, the relative methylene groups yielded two close but distinct triplets at 4.67 (*22*) and 4.75 (*27*) ppm. In the downfield region, two doublets relative to the aromatic portion of the tosylate anion are visible at 7.21 and 7.68 ppm. The azobenzene unit’s substitution pattern introduces magnetical differences in the aromatic protons *ortho* to the alkoxy substituents, labeled as (*13*) and (*16*) in Figure 1, as well as those *ortho* to the azo group, labeled as (*14*) and (*15*). Each pair of coupled protons gives rise to two overlapping doublets: protons (*13*) and (*16*) at 7.02 ppm and protons (*14*) and (*15*) at 7.82 ppm. These patterns are nearly identical in both spectra (Figure 2). The most significant difference between the spectra of **11** and **10** lies in the pattern of signals arising from the two pyridine aromatic rings of the V-station. The higher symmetry of this system in **10** results in two overlapped doublets at approximately 9.2 ppm for the protons *ortho* to the positively charged nitrogen (*23*) and (*26*). The less deshielded protons (*24*) and (*25*) resonate as a doublet at 8.62 ppm. In contrast, the asymmetry of the V-station in semi-axle **11**, which is alkylated on only one pyridine ring, generates a distinct pattern of four doublets at 9.10, 8.79, 8.46, and 7.94 ppm, corresponding to protons (*23*), (*26*), (*24*), and (*25*), respectively.

### 2.2. Synthesis of Rotaxane **12**(Azo-Up)

In low polarity solvents, such as chloroform, dichloromethane, or toluene, the threading in calix[6]arene host such as **TPU** can be directionally controlled from the macrocycle upper rim by using mono-stoppered viologen axles [31]. The equilibration of **10** with **TPU** in dichloromethane at room temperature resulted in the formation of a red color solution. This characteristic color, usually arising from the charge transfer interaction between the electron-deficient axle and the electron-rich calixarene cavity, confirmed the formation of a pseudorotaxane complex [29]. The subsequent stoppering-protection reaction was performed directly on the same solution using TBSCl, with triethylamine as the base and DMAP as the catalyst. This reaction yielded the expected orientational rotaxane isomer **12**(*azo-up*) in 47% yield (Figure 2). The resulting compound, a MIM, features, in fact, two bulky TBS protecting groups that effectively prevent the viologen-based dumbbell from slipping out from the macrocycle. Indeed, **12**(*azo-up*) was successfully purified and isolated via column chromatography. The identity of this rotaxane and the reciprocal orientation of its two molecular components were confirmed through a series of detailed NMR measurements (see Appendix A). These results also provided indirect confirmation of the selective formation of the pseudorotaxane intermediate **TPU**⊃**10**(*azo-up*) during the threading reaction.

The ^1^H NMR spectrum of rotaxane **12**(*azo-up*), recorded in CD_2_Cl_2_, has been reported in Figure 4b. In comparison to the free **TPU** (Figure 4a), **12**(*azo-up*) shows a more resolved pattern of signals, indicating an increase in the rigidity of the macrocycle due to the interlocking of its molecular components (cf. Figure 4a,b). This hypothesis is further supported by the sharpening and downfield shift of the signal for the macrocycle methoxy groups from 2.9 to 3.98 ppm in the rotaxane. This downfield shift occurs because the methoxy groups are ejected from the cavity upon axle threading and thus resonate at a lower field. Other diagnostic signals confirming the threading and blocking of axle **10** inside the cavity of **TPU** include (i) the significant upfield shift of the bispyridinium proton signals, which move from 9.3–8.6 (Figure 4a) in **10** to ca. 7.7–5.9 ppm in the rotaxane (cf. Figure 4b,c). This upfield shift is due to the pyridinium rings becoming more shielded inside the aromatic cavity of **TPU**, resulting in broader signals. (ii) The appearance of two broad signals for the NH protons of the macrocycle’ phenylureido groups at 8.9 and 8.6 ppm. These protons are hydrogen-bonded to the tosylate counteranions of the threaded axle. (iii) The upfield shift of the methylene protons (*22*) and (*27*) from approximately 4.8 and 4.7 to 3.2 and 3.9 ppm, respectively. In the spectrum of **12**(*azo-up*), these resonances become very broad, and their precise locations were determined through the analysis of an HSQC spectrum (see Appendix A). The different shift of these two resonances (ca. 1.6 and 0.8 ppm) is attributed to the known geometry of inclusion of the bispyridinium core (V-station) of this type of viologen axle in calix[6]arene macrocycles like **TPU** [36,40]. Indeed, the axle methylene group (*27*) protrudes from the lower rim of the **TPU** cavity. Conversely, the opposite methylene (*22*) is located on the cavity’s upper rim, experiencing a large anisotropic shielding effect, also thanks to the three phenylureas decorating the **TPU** cavity. It is also noteworthy that the two singlets, due to the *tert*-butyl and methyl groups of the TBS protecting groups, are split because of the asymmetry of the rotaxane. One TBS group is close to the macrocycle upper rim, while the other is close to the lower rim. Analogously, the adjacent methylene groups, labeled (*1*) and (*32*), are split into two resonances now at 3.59 ppm (*1*) and 3.79 ppm (*32*). Very important for the identification of the signals was the identification of the 2D TOCSY correlations between (*22*) and (*17*), (*1*) and (*6*), and (*27*) and (*32*). Moreover, the signals (*22*) and (*17*) have TOCSY correlations with the four upfield shifted methylene groups resonating at 0.52 ppm (*21*), 0.84 ppm (*20*), 1.86 ppm (*19*), and 1.66 ppm (*18*). These findings confirm that this portion of the guest is near the phenylureido groups at the upper rim of **TPU** and is strongly influenced by its shielding effect (see Appendix A).

### 2.3. Synthesis of Rotaxane **12**(Azo-Down)

As briefly introduced in the previous paragraph, the synthesis of rotaxane **12**(*azo-down*) (see Figure 3) could not be achieved using the threading-and-capping strategy previously employed for its counterpart, **12**(*azo-up*). This approach would have required threading from the macrocycle upper rim using an axle component bearing (i) a TBS stopper on the terminus of the short alkyl chain linked to the V-station; (ii) a free hydroxyl group on the other terminus of the axle, and (iii) the removal of the Boc protecting group prior to performing the threading reaction. However, synthesizing an axle component with these specific features is inherently long and complicated. Moreover, the outcome of the unidirectional threading of such an axle cannot be reliably predicted. Given these challenges, we preferred to adopt the active template approach, which we previously used successfully for the synthesis of other calix[6]arene-based rotaxanes. This method, which we termed “supramolecular-assisted alkylation” [32,41,42], is based on the in situ formation of the dumbbell within the calix[6]arene cavity, starting from a mono-stoppered pyridylpyridinium semi-axle and a stoppered alkylating agent. Notably, this method consistently produces high orientational selectivity when asymmetric dumbbells must be synthesized. Specifically, the alkylation reaction ensures that the resulting [n]rotaxane (n = 2 or 3) always presents the stopper originally present on the semi-axle at the calix[6]arene lower rim.

To implement the supramolecular-assisted alkylation to our aims, the mono-stoppered TBS-protected semi-axle **11** was prepared in 67% yield by alkylating 4,4′-bipyridyl with tosylate **8** in refluxing acetonitrile for 48 h (Figure 1). The semi-axle **11** was then equilibrated with **TPU** in dry toluene for 24 h at a 1:1.5 molar ratio, followed by a reaction with a stoichiometric amount of 6-hydroxyhexyltosylate **13** for a further seven days (Figure 3). The formation of pseudorotaxane species was witnessed by a color change in the reaction mixture from pale orange to reddish. The resulting pseudorotaxane isomer **TPU**⊃**10**(*azo-down*) was not isolated but directly converted into the rotaxane **12**(*azo-down*) by stoppering/protecting the hydroxyl terminus of the complexed **10** with TBSCl (see Figure 3). Rotaxane **12**(*azo-down)* was isolated as a red solid compound with an overall yield of 47% following chromatographic separation. This compound was fully characterized by MS and NMR analysis (see also Appendix A).

The ^1^H NMR spectra of the two orientational rotaxane isomers **12**(*azo-down*) and **12**(*azo-up*) are presented in the stack plot of Figure 5 for comparison. While the overall signal patterns in the two spectra are very similar, a few notable differences are observed: (i) the chemical shifts of the resonances assigned to the dumbbell’s bispyridinium unit (V-station) protons (*23–26*) (see labels in Figure 2 and Figure 3) are interchanged. This supports the hypothesis that the dumbbell orientation within the macrocycle differs between the two rotaxanes. Additionally, in the spectrum of rotaxane **12**(*azo-up*), the signals corresponding to the methylene protons (*12*) and (*17*), which are connected to the azobenzene unit, are slightly upfield-shifted (approximately 0.2 ppm) compared to those in **12**(*azo-down*). This finding agrees with the fact that in **12**(*azo-up*), the azobenzene unit resides over the calix[6]arene upper rim, where the protons still experience the anisotropic shielding effect of the aromatic cavity. Since the HR-MS spectra of both compounds exhibit identical molecular ions (see Appendix A), these findings collectively confirm the successful preparation of both rotaxane orientational isomers.

### 2.4. Synthesis of the Catenane

The synthesis of catenanes starting from the previously synthesized rotaxane isomers requires addressing two key challenges: (i) developing a suitable method to liberate the thread termini without destabilizing the resulting pseudorotaxane complex, which should serve as the precursor in the catenation reaction; and (ii) identifying an appropriate “clipping unit”. This latter must be a difunctional compound capable of reacting rapidly with the re-established hydroxyl groups on the thread and functioning, ideally, as a gate (G2) with photophysical properties orthogonal to the azobenzene unit (G1) already installed on the ring track. Based on the *E*→*Z* and *Z*→*E* photoisomerization wavelengths of 4,4′-dimethoxy azobenzene (model compound for G1), which occur at 365 and 465 nm [43], respectively, we selected the *trans*-stilbene-4,4′-dicarboxylic acid **15** (Figure 4) as a suitable candidate for G2. The literature reports indicate that the *Z*→*E* photoisomerization process of a *trans*-stilbene-dicarboxylic acid-(4.4′)-diamide derivative occurs at 350 nm, while the reverse *Z*→*E* process occurs at 280 nm [44]. Therefore, G2 can remain in its *H* state when G1 is returned to its *NH* state. Additionally, upon its activation to an acyl chloride, **15** allows, in principle, rapid esterification with the thread hydroxyl termini under mild conditions, avoiding the pseudorotaxane destabilization risks posed by base-catalyzed reactions.

Compound **16** was synthesized via the two-step procedure outlined in Figure 4. Its precursor **15** was synthesized in 73% yield starting from the commercially available 4-bromo benzoic acid **14** and triethoxyethyl vinyl silane in a Heck/Hiyama coupling reaction catalyzed by Pd(OAc)_2_ [45]. The carboxylic groups of **15** were then converted into more reactive acyl chloride groups to enable faster and milder clipping reactions. This transformation was achieved using thionyl chloride in DMF at 70 °C for 24 h, quantitatively yielding the acyl chloride **16** as a yellow solid.

The catenation reaction was initially attempted on the pseudorotaxane orientational isomers **TPU**⊃**10**(*azo-up*) and **TPU**⊃**10**(*azo-down*). Unfortunately, these efforts were unsuccessful due to the challenges in removing the TBS protecting groups on the thread termini. Specifically, the tetrabutylammonium fluoride (TBAF) and KF used in the reactions compromised the stability of the pseudorotaxane complexes, either by affecting the phenylurea moieties of the macrocycle or disrupting the V-station of the thread.

To address the above issues, a supramolecular-assisted approach was adopted to synthesize the precursor of the catenane track ring, i.e., the thread with the free hydroxyl groups, directly within the cavity of the macrocycle but with the desired geometrical arrangement. This strategy was based on the idea that the Boc group on the N-station was sufficiently bulky to guide the orientation of the newly formed dumbbell as found for rotaxane **12**(*azo-down*). To this aim, a new semi-axle, compound **18**, was prepared starting from tosylate **8** (Figure 4). Its protecting group (TBS) was removed using CuCl_2_ in a water/acetone mixture, a mild deprotection method that preserves the integrity of both the azobenzene and the Boc of **8**. The resulting compound **17** was then reacted with 4,4′-bipyridine in refluxing acetonitrile to afford semi-axle **18** with a 59% yield.

The oriented catenane **C3**(*azo-down*) was synthesized using a procedure analogous to that employed for the preparation of rotaxane **12**(*azo-down*). It is crucial to conduct the catenation reaction in a weakly polar medium to prevent scrambling of the thread within the pseudorotaxane complex and to ensure the correct orientation and functionality are retained. As a result, in the first step, pseudorotaxane **TPU**⊃**19**(*azo-down*) was prepared by mixing **TPU** and **18** in a 1.5:1 stoichiometric ratio in dry toluene. The mixture was stirred at 60 °C for 24 h, and then tosylate **13** was added (Figure 4). The reaction was then continued at 70 °C until a persistent red coloration of the mixture was observed (usually seven days). The oriented pseudorotaxane containing the hypothetical thread **19** (see Figure 4) was not isolated. Instead, the toluene solvent was distilled, and the resulting residue was dissolved in dichloromethane to obtain a reddish ca. 10^−3^ M solution containing the presumed oriented pseudorotaxane **TPU**⊃**19**(*azo-down*). As seen in previous studies [35,36], this concentration should minimize possible intermolecular reactions. To this solution, a mixture of bisacyl chloride **16**, *N*,*N*-dimetylaminopyridine (DMAP) and triethylamine in the same solvent was added. The reaction proceeded to completion, affording the desired catenane in a 25% yield following chromatographic purification.

The identity of **C3**(*azo down*) was initially confirmed through mass spectroscopy and NMR analyses. The high-resolution mass spectrum (ESI-ORBITRAP) revealed the presence of a doubly charged base peak at *m*/*z* = 1275.71559, attributed to the loss of the two tosylate anions from the MIM (Figure 6). To verify the geometrical orientation of the track ring encapsulated within the **TPU** macrocycle, the ^1^H NMR spectrum of **C3**(*azo-down*) was compared with that of the rotaxane **12**(*azo-down*), which shares the same geometrical arrangement of the azobenzene unit (G1) relative to the macrocycle rims with the catenane (Figure 7). Integration of the low-field region of the spectra confirmed the successful incorporation of the stilbene unit (G2) into the track ring. It should be observed that **C3** lost its tosylates during the purification step. The resonances observed at δ > 8 ppm were assigned to the aromatic protons of the stilbene moiety. Notably, the increased spectral complexity of **C3**(*azo-down*) compared to **12**(*azo-down*) is likely a consequence of the cyclization reaction, which introduces additional asymmetry into the molecular system. For instance, the bridging methylene groups of the calix[6]arene macrocycle in **C3** do not exhibit the characteristic AX pattern of two doublets with a geminal coupling that is observed for the equatorial and axial diastereotopic protons in **12**(*azo-down*).

In conclusion, this study demonstrates the successful design and synthesis of a calix[6]arene-based [2]catenane incorporating two distinct gate units to achieve orthogonal photophysical responses. The supramolecular-assisted strategy proved crucial for overcoming synthetic challenges and ensuring precise molecular orientation. The findings establish catenane C3(*azo-down*) as a promising prototype for advanced rotary molecular motors, with the potential for controlled directional motion mediated by photoisomerizable gates. In fact, since the V-station is redox-active and its interaction with **TPU** is disrupted upon reduction [29], we aim to investigate whether electrochemical stimulation of **C3**(*azo-down*) induces co-conformational changes. While the development of a rotary motor is beyond the scope of this study, the desymmetrized structure of catenanes like **C3**(*azo-down*) makes them promising candidates for exploring directional rotation mechanisms. Future work will explore the dynamic behavior and functional capabilities of this two-gate system under external stimuli.

## 3. Materials and Methods

### 3.1. General

All solvents were dried using standard procedures; all other reagents were of reagent-grade quality, obtained from commercial suppliers, and were used without further purification. NMR spectra were recorded at 300 MHz and 400 MHz Bruker AVANCE or at 600 MHz JEOL ECZ600R for ^1^H and 100 MHz or at 150 MHz for ^13^C. Melting points were recorded on an Electrothermal 9100 apparatus and were uncorrected. Chemical shifts are expressed in ppm (δ) using the residual solvent signal as an internal reference (7.26 ppm for CHCl_3_, 5.32 for CHDCl_2_, and 3.31 for CD_2_HOD). Mass spectra were recorded employing an Infusion Water Acquity Ultra Performance LC HO6UPS-823M mass spectrometer and an LTQ ORBITRAP XL Thermo with an ESI source (electrospray) in positive mode. Compounds **1** [46], **2** [47], **4** [39], **5** [48], **6** [49], **9** [40], **14** [50], and the stilbene derivatives **15** [45] and **16** [51] were synthesized according to published procedures.

### 3.2. Synthetic Procedures

***tert*-Butyl(6-bromohexyl)(6-((*tert*-butyldimethylsilyl)oxy)hexyl) carbamate (3)** NaH (60% dispersion in mineral oil, 0.16 g, 6.67 mmol) was added to a solution of **2** (1.0 g, 3.02 mmol) in 20 mL of dry THF and the mixture was stirred at r.t. under nitrogen flux for 1 h. 1,6-dibromohexane (1.38 mL, 9.06 mmol) was added and the mixture was stirred for 4 h. The reaction was quenched by the addition of iced distilled water (200 mL), and the resulting mixture was extracted with CH_2_Cl_2_ (150 mL). The organic layer was washed with distilled water 3 times (3 × 150 mL), dried on anhydrous Na_2_SO_4_, and evaporated under reduced pressure. The oily crude obtained was purified by column chromatography (SiO_2_, n-hexane/EtOAc 9:1) to give 0.90 g of **3** as a colorless oil in a 60% yield. ^1^H NMR (CDCl_3_, 400 MHz, 298 K): δ (ppm) = 3.56 (t, 2H, *^3^J* = 6.5 Hz), 3.36 (t, 2H, *^3^J* = 6.8 Hz), 3.17–3.07 (br.s, 4H), 1.86–1.79 (m, 2H), 1.53–1.46 and 1.42 (m, s, 17H), 1.35–1.20 (m, 6H), 0.86 (s, 9H), 0.01 (s, 6H). ^13^C NMR (CDCl_3_, 100 MHz, 298 K): δ (ppm) = 155.7, 79.0, 63.2, 47.1, 47.0, 33.7, 32.9, 32.8, 28.6 (two resonances), 28.0, 26.8, 26.1, 26.0 (two resonances), 18.4, 5.2. ESI-MS (+) calculated for [C_23_H_48_BrNO_3_Si+Na^+^] *m*/*z* = 518, found for [M+Na^+^] 518; calculated for [C_23_H_48_BrNO_3_Si + K^+^] *m*/*z* = 534, found for [M+K^+^] 534.

***tert*-butyl (*E*)-(6-((*tert*-butyldimethylsilyl)oxy)hexyl)(6-(4-((4-(6-hydroxyhexyloxy)phenyl)diazenyl)phenoxy)hexyl)carbamate** (**7**) K_2_CO_3_ (0.31 g, 2.24 mmol) was added to a solution of 6 (0.58 g, 1.85 mmol) in 30 mL of dry ACN and the mixture was stirred at r.t. for 10 min. Compound **3** (1 g, 2.02 mmol) and KI (cat.) were added and the reaction was refluxed 48 h. ACN was evaporated under reduced pressure and the crude was collected with EtOAc (100 mL). The resulting organic layer was treated with a 10% *w*/*v* solution of HCl until neutrality and then washed twice with distilled water (2 × 150 mL), dried on anhydrous Na_2_SO_4_, and evaporated under reduced pressure. The crude was purified by column chromatography (SiO2, n-hexane/EtOAc 1:1) to give 1.05 g of 7 as a yellow solid in a 78% yield. m.p. = 70.8–71.1 °C. ^1^H NMR (CDCl_3_, 300 MHz, 298 K): δ (ppm) = 7.86 (d, 4H, *^3^J* = 8.9 Hz), 6.98 (d, 4H, *^3^J* = 7.8), 4.07–3.98 (m, 4H), 3.67 (t, 2H, *^3^J* = 6.5 Hz), 3.59 (t, 2H, *^3^J* = 6.4 Hz), 3.15 (br.s, 4H), 1.87–1.78 (m, 4H), 1.65–1.48, 1.39–1.25 and 1.45 (2m, s, 37H), 0.89 (s, 9H), 0.04 (s, 6H). ^13^C NMR (CDCl_3_, 100 MHz, 298 K): δ (ppm) = 162.0, 146.6, 124.5, 116.3, 79.9, 68.9, 63.3, 62.2, 47.0, 33.6, 32.8, 29.3, 28.6, 27.3, 26.8 (two resonances), 26.1, 26.0, 25.8, 25.7, −5.1. ESI-MS(+) calculated for [C_41_H_69_N_3_O_6_Si + H^+^] *m*/*z* = 728, found for [M + H^+^] 728.

**(*E*)-6-(4-((4-((6-((*tert*-butoxycarbonyl)-(6-((*tert*-butyldimethylsilyl)oxy)hexyl)amino)hexyl)oxy)phenyl)diazenyl)phenoxy)hexyl 4-methylbenzenesulfonate** (**8**) Triethylamine (0.16 mL, 1.57 mmol) and DMAP (cat.) were added to a solution of 7 (0.90 g, 1.24 mmol) in 30 mL of CH_2_Cl_2_. The mixture was cooled at 0 °C, 4-methylbenzenesulfonylchloride (0.25 g, 1.30 mmol) was added, and the reaction was stirred at r.t. for 24 h. The organic layer was washed twice with distilled water (2 × 100 mL), dried on anhydrous Na_2_SO_4_, and evaporated under reduced pressure. The crude was purified by column chromatography (SiO_2_, n-hexane/EtOAc 9:1) to give 0.97 g of 8 as a yellow solid in 89% yield. m.p. = 81.9–83.2 °C. ^1^H NMR (CDCl_3_, 300 MHz, 298 K): δ (ppm) = 7.87 (d, 4H, *^3^J* = 8.8 Hz), 7.79 (d, 2H, *^3^J* = 8.2 Hz), 7.34 (d, 2H, *^3^J* = 7.4 Hz), 6.98 and 6.95 (2d, 4H, *^3^J* = 8.8 Hz), 4.04 (q, 4H, *^3^J* = 6.4 Hz), 3.99 (t, 2H, *^3^J* = 6.4 Hz), 3.59 (t, 2H, *^3^J* = 6.4 Hz), 3.20–3.10 (br.s, q, 4H), 2.44 (s, 3H), 1.87–1.67 (2m, 6H), 1.55–1.22 and 1.45 (m and s, 30H), 0.89 (s, 9H), 0.04 (s, 6H). ^13^C NMR (CDCl_3_, 150 MHz, 298 K): δ (ppm) = 147.2, 144.8, 144.3, 136.3, 130.0, 128.0, 124.5, 122.9, 114.8 (two resonances), 79.1, 70.6, 68.3, 68.1, 63.4, 47.1, 33.0, 29.4, 29.1, 28.9, 28.7, 26.9, 26.1, 26.0, 25.8, 25.6, 25.3, 21.8. ESI-MS(+) calculated for [C_48_H_75_N_3_O_8_SSi + H^+^] *m*/*z* = 882, found for [M + H^+^] 882.

**Axle (10).** Compound **8** (0.21 g, 0.23 mmol) was added to a solution of **9** (0.10 g, 0.23 mmol) in 10 mL of dry acetonitrile, and the mixture was refluxed for 7 days. The solvent was evaporated under reduced pressure, and the crude was purified by recrystallization from ACN, affording 0.17 g of **10** as an orange solid in a 55% yield. m.p. = 180.5–184.3 °C. ^1^H NMR (CD_3_OD, 300 MHz, 298 K): δ (ppm) = 9.24 and 9.21 (2d, 4H, *^3^J* = 6.7 and 6.5 Hz), 8.62 (d, 4H, *^3^J* = 6.0 Hz), 7.83 and 7.81 (2d, 4H, *^3^J* = 7.0 and 6.9 Hz), 7.68 (d, 4H, *^3^J* = 8.1 Hz), 7.21 (d, 4H, *^3^J* = 8.1 Hz), 7.03 (t, 4H, *^3^J* = 9.3 Hz), 4.75 (t, 2H, *^3^J* = 7.6 Hz), 4.67 (t, 2H, *^3^J* = 7.6 Hz) 4.09–4.05 (m, 4H), 3.57–3.52 (m, 4H), 3.20 (br.s, q, 4H), 2.35 (s, 6H), 2.15–2.01 (m, 4H), 1.87–1.78 (m, 4H), 1.66–1.29 and 1.45 (m and s, 32H), 0.91 (s, 9H), 0.07 (s, 6H). ^13^C NMR (CD_3_OD, 100 MHz, 298 K): δ (ppm) = 148.2, 147.1, 141.7, 129.9, 129.8, 128.3, 128.2, 126.9, 125.3, 125.2, 115.8, 69.3, 63.2, 62.9, 62.6, 33.6, 33.2, 32.4, 30.3, 28.8, 27.6, 26.9, 26.7, 26.4, 26.4, 21.3. ESI-MS(+) calculated for [C_57_H_89_N_5_O_6_Si^2+^] *m*/*z* = 484 found for [M^2+^ ‒ 2TsO^−^] 484.

**Semi-axle (11).** Compound **8** (0.30 g, 0.34 mmol) and 4,4′-bipyridyl (0.05 g, 0.35 mmol) were dissolved in 20 mL of dry acetonitrile. The resulting reaction mixture was refluxed for 48 h. After this period, the solvent was evaporated under reduced pressure. The crude residue was purified by precipitation from ethyl acetate to give 0.24 g of **11** as a yellow solid in a 67% yield. m.p. = 116.6–117.2 °C. ^1^H NMR (300 MHz, CDCl_3_): δ (ppm) = 9.05 (d, 2H, *^3^J* = 6.7 Hz), 8.79 (d, 2H, *^3^J* = 5.9 Hz), 8.44 (d, 2H, *^3^J* = 6.7 Hz), 7.93 (d, 2H, *^3^J* = 6.1 Hz), 7.84 (d, 2H, *^3^J* = 8.6 Hz), 7.82 (d, 2H, *^3^J* = 8.6 Hz), 7.7 (d, 2H, *^3^J* = 8.1 Hz); 7.21 (d, 2H, *^3^J* = 8.1 Hz), 7.04 (d, 2H, *^3^J* = 8.4 Hz), 7.02 (d, 2H, *^3^J* = 8.4 Hz), 4.68 (t, 2H, *^3^J* = 7.4 Hz), 4.07 (t, 4H, *^3^J* = 6.2 Hz), 3.64 (t, 2H, *^3^J* = 6.2 Hz), 3.21 (q, 4H, *^3^J* = 6.8 Hz), 2.37 (s, 3H), 2.16–2.01 (m, 2H), 1.9–1.74 (m, 4H), 1.69–1.22 and 1.47 (m and s, 29H), 0.91 (s, 9H), 0.06(s, 6H).

**Rotaxane (12(*azo-up*)). TPU** (0.04 g, 0.02 mmol) was added to a solution of **10** (0.03 g, 0.02 mmol) in 5 mL of dry CH_2_Cl_2_ and the solution was stirred at r.t. for 2 h. Then, TBSCl (0.004 g, 0.03 mmol), triethylamine (0.004 mL, 0.03 mmol), and DMAP (cat.) were added and the reaction mixture was stirred at r.t. for 24 h. The organic layer was washed twice with distilled water (2 × 50 mL), dried on anhydrous Na_2_SO_4_, and evaporated under reduced pressure. The crude was purified by column chromatography (SiO_2_, CH_2_Cl_2_/MeOH 96:4) to give 0.03 g of **12**(*azo-up)* as a reddish solid in a 47% yield. ^1^H NMR (CDCl_3_, 300 MHz, 298 K): δ (ppm) = 8.93 (s, 3H), 8.55 (s, 3H), 7.88 (m, 4H), 7.76 (d, *^3^J* = 8.3 Hz, 4H), 7.46–7.29 (m, 23H), 7.19–7.03 (m, 8H), 7.00 (d, *^3^J* = 9.0 Hz, 4H), 6.83 (d, *^3^J* = 7.5 Hz, 8H), 6.73 (d, *^3^J* = 6.2 Hz, 2H), 6.00 (d, *^3^J* = 6.1 Hz, 2H), 4.49 (d, *^2^J* = 15.0 Hz, 6H), 4.26 (br.t, 2H), 4.04 (br.t, 2H), 3.99 (t, *^3^J* = 3.3 Hz, 9H), 3.84 (br. s, 4H), 3.69–3.42 (m, 10H), 3.15 (br.q., 6H), 2.44 (s, 6H), 1.92–1.13, 1.49 and 1.43 (m and 2s, 39H), 0.88 (s, 18H), 0.04 (s, 12H). ^13^C NMR (CD_2_Cl_2_, 100 MHz, 298 K): δ (ppm) = 143.93, 142.89, 130.20, 128.91, 127.72, 124.91, 123.59, 121.46, 116.96, 115.14, 114.15, 72.71, 70.68, 69.92, 68.26, 67.98, 66.51, 63.00, 62.79, 62.42, 61.43, 61.07, 60.52, 46.83, 32.79, 32.26, 31.22, 29.14, 28.70, 28.15, 26.58, 25.65, 24.70, 20.81, 14.55, −5.61. HR-MS calculated for [C_153_H_211_N_11_O_18_Si_2_]^2+^ *m*/*z* = 1273.2731 (60%), 1273.7747 (100.0%), 1274.2764 (82%), 1274.7781 (26%); Found: 1273.2693 (55%), 1273.7710 (100%), 1274.2721 (90%), 1274.7730 (60%), 1275.2745 (30%).

**Rotaxane (12(*azo-down*)). TPU** (0.26 g, 0.18 mmol) was added to a solution of **12** (0.12 g, 0.12 mmol) in 5 mL of dry toluene and the solution was stirred at 60 °C for 24 h. Then, tosilate **13** (0.033 g, 0.12 mmol) was added and the reaction was stirred at 70 °C for 7 days. After that, TBSCl (0.044 g, 0.29 mmol), triethylamine (0.04 mL, 0.29 mmol), and DMAP (cat.) were added and the reaction mixture was stirred at r.t. for 24 h. The solvent was then evaporated under reduced pressure and the crude was collected with CH_2_Cl_2_ (50 mL). The organic layer was washed twice with distilled water (2 × 50 mL), dried on anhydrous Na_2_SO_4_, and evaporated under reduced pressure. The crude was purified by column chromatography (SiO_2_, CH_2_Cl_2_/MeOH 95:5) to give 0.18 g of **12**(*azo-down)* as a reddish solid in a 47% yield. ^1^H NMR (CDCl_3_, 300 MHz, 298 K): δ (ppm) = 8.88 (br.s, 3H), 8.61 (br.s, 3H), 7.94–7.84 (m, 8H), 7.77 (d, *^3^J* = 8.0 Hz, 2H), 7.37 (br.s, 21H), 7.22 (d, *^3^J* = 8.0 Hz, 2H), 7.15–6.97 (m, 6H), 6.82 (d, 9.7 Hz, 4H), 5.87 (d, *^3^J* = 6.2 Hz, 4H), 4.48 (d, *^2^J* = 15.0 Hz, 6H), 4.08–3.96 (m, 11H), 3.86 (br. m, 2H), 3.62 (m, 22H), 3.47 (d, *^2^J* = 15.1 Hz, 6H), 3.15 (m, 6H), 2.38 (s, 6H), 1.93–1.18, 1.48 and 1.43 (m and 2s, 41H), 0.91 (s, 18H), 0.08 (s, 12H). ^13^C NMR (CD_2_Cl_2_, 100 MHz, 298 K): δ (ppm) = 144.4, 143.4, 131.6, 126.4, 125.2, 124.2, 122.7, 118.7, 116.7, 115.1, 73.3, 70.5, 68.8, 68.4, 67.1, 63.5, 63.4, 61.9, 60.9, 47.4, 34.1, 33.3, 31.2, 29.7, 29.5, 28.7, 27.2, 26.2, 21.5, 15.0, −5.08. HR-MS calculated for [C_153_H_211_N_11_O_18_Si_2_]^2+^ *m*/*z* = 1273.2731 (60%), 1273.7747 (100.0%), 1274.2764 (82%), 1274.7781 (26%); Found: 1273.2716 (55%), 1273.7731 (100.0%), 1274.2749 (90%), 1274.7760 (60%), 1275.2770 (30%).

***tert*-butyl (*E*)-(6-((*tert*-butyldimethylsilyl)oxy)hexyl)(6-(4-((4-((6-hydroxyhexyl)oxy)phenyl)diazenyl)phenoxy)hexyl) carbamate** (**17**) CuCl_2_ (0.03 mmol) was added to a solution of 8 (0.30 g, 0.34 mmol) in 10 mL of a 19:1 mixture of acetone/water. The mixture was heated at 50 °C for 2 h. The crude reaction mixture was filtered on silica gel and evaporated under reduced pressure, affording 0.25 g of **17** in quantitative yield. The product was immediately used without further purification.

**Semi-axle (18).** Compound **17** (0.12 g, 0.15 mmol) and 4,4‘-bipyridyl (0.04 g, 0.23 mmol) were dissolved in 20 mL of dry acetonitrile. The resulting reaction mixture was refluxed for 48 h. After this period, the solvent was evaporated under reduced pressure. The crude residue was purified by precipitation from ethyl acetate to give 0.08 g of **18** as a yellow solid in a 59% yield. ^1^H NMR (CDCl_3_, 300 MHz, 298 K): δ (ppm) = 9.09 (d, 2H, ^3^J = 6.8 Hz), 8.79 (d, 2H, ^3^J = 6.0 Hz), 8.45 (d, 2H, ^3^J = 9.8 Hz), 7.95 (d, 2H, ^3^J = 5.8 Hz), 7.86–7.81 (m, 4H), 7.70 (d, 2H, ^3^J = 8.2 Hz); 7.22 (d, 2H, ^3^J = 8.0 Hz), 7.06–7.01 (m, 4H), 4.68 (t, 2H, ^3^J = 7.4 Hz), 3.24–3.18 (m, 4H), 2.34 (s, 3H), 2.13–2.08 (m, 2H), 1.88–1.81 (m, 4H), 1.70–1.20 (m, 27H). ESI-MS(+): 752 [M^+^ ‒ TsO^−^].

**Catenane C3(*azo-down*). TPU** (0.36 g, 0.24 mmol), **18** (0.15 g, 0.16 mmol), and **13** (0.05 g, 0.18 mmol) were equilibrated in 3 mL of toluene. The reaction was heated at 70 °C for 7 days in a sealed glass autoclave. After evaporation of the solvent, the mixture was taken up with dry CH_2_Cl_2_ (200 mL) to obtain a ≈10^−3^ M solution. To this solution, triethylamine (0.032 g, 0.32 mmol) and DMAP (cat.) were added. To the resulting reaction mixture, a solution of **16** (0.048 g, 0.16 mmol) in dry CH_2_Cl_2_ was added dropwise (10 min). The reaction was stirred at room temperature for 72 h. After this period, the solvent was evaporated to dryness under vacuum. The reddish crude solid residue was purified by chromatography (SiO_2_, CH_2_Cl_2_/MeOH 95:5) to afford 0.12 g of **C3**(*azo down*) in a 25% yield. For the ^1^H NMR spectrum, see Appendix A. The asymmetry of the resulting MIM prevents the ^13^C NMR analysis. HR-MS calculated for [C_157_H_191_N_11_O_20_]^2+^ *m*/*z* = 1275.2128 (59%), 1275.7145 (100.0%), 1276.2161 (83%), 1276.7178 (46%), 1277.2195 (20%), 1277.7212 (8%); Found: 1275.2138 (59%), 1275.7155 (100.0%), 1276.2168 (91%), 1276.7179 (53%), 1277.2192 (23%), 1277.7207 (9%).

## Data Availability

Data is contained within the article.

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
