# Peer review of "Synthesis and Characterization of a Two-Station Two-Gate Calix[6]arene-Based [2]Catenane"

_molecules, 2025, doi:10.3390/molecules30030732_

Round 1
Reviewer 1 Report
Comments and Suggestions for Authors
In the manuscript, the authors demonstrated the design and synthesis of a ca-lix[6]arene-based [2]catenane with azobenzene and stilbene moieties. The supramolecular-assisted strategy was used to synthesize the products. Mass spectrometry and NMR spectroscopy confirmed the successful synthesis and orientation of C3(azo-down). Overall, the conclusion can be supported the experimental results, and is credible.
Author Response
No actions required by Reviewer 1.
Reviewer 2 Report
Comments and Suggestions for Authors
This paper submitted by Secchi et al. describes the preparation of a calix[6]arene-based catenane comprising two stations and two gates at second ring. Initially, the authors managed to synthesize and characterize two rotaxanes, which are orientational isomers between them. The obtention of the corresponding catenanes, by an initial deprotection step followed a clipping reaction was unsuccessful. By changing the synthetic route, they finally obtained one of the two isomeric desired catenane. Although the authors isolated a very interesting photoresponsive system, they did not perform further experiments in order to control the relative position of the calix[6]arene ring along the second macrocycle by the applications of external stimuli.
In my opinion, the article should be published once the authors have considered the corrections suggested hereafter:
The molecules reported in the article are complex, and the authors fully characterized them with bidimensional NMR experiments. This is the case for all the compounds, except for the most important one in this work, catenane C3. There is no 1H signal assignment such as in the rest of the molecules and, in addition, the authors claimed that the asymmetry of compound C3 prevents the 13C analysis. In my opinion the asymmetry of compound C3 is similar to that of rotaxanes 12, although with a higher motion restriction. Thus, a complete NMR analysis must be done for C3, to unequivocally characterize it. If not, in my opinion, this work is not suitable for publishing.
Points at the main text:
1) Line 279: it should be Scheme 1 instead of Scheme 2.
2) Scheme 3: at the left part, the first molecule should be labelled as 11, not 12.
3) For clarity, I think it is better to separate Scheme 4 in two parts (a and b). Part a: Synthesis of trans-stilbene derivative 16. Part b: As in the article.
4) Figure 4. The 1H NMR of axle 10 is reported in MeOD, while in the line 468 is written CDCl3. This error should be fixed.
Points at the supplementary material:
Compounds 8 and axle 10: only the 13C spectra are reported. In addition, the spectrum of axle 10 has a poor quality. I suggest to replace these spectra for APT spectra of good quality.
Author Response
Reviewer 2 general comment: Although the signals' resolution of the proton spectrum of catenane C3 was overall poor, because of the lack of symmetry of such MIM, we provided a tentative proton assignment based on the interpretation of several 2D NMR experiments.
Specific requests:
Q1. Line 279: it should be Scheme 1 instead of Scheme 2.
A1. Fixed
Q2. Scheme 3: at the left part, the first molecule should be labelled as 11, not 12.
A2. Corrected
Q3. For clarity, I think it is better to separate Scheme 4 in two parts (a and b). Part a: Synthesis of trans-stilbene derivative 16. Part b: As in the article.
A3. We thank the referee for this suggestion. We prepared a new, improved, Scheme 4.
Q4. Figure 4. The 1H NMR of axle 10 is reported in MeOD, while in the line 468 is written CDCl3. This error should be fixed.
Q4. Fixed
Q-Supplementary Material. Compounds 8 and axle 10: only the 13C spectra are reported. In addition, the spectrum of axle 10 has a poor quality. I suggest to replace these spectra for APT spectra of good quality.
A-SM. We replaced the spectra as suggested by recording DEPT-Q experiments instead of APT. In general, we corrected and expanded (mass spectra) all the SM.
Reviewer 3 Report
Comments and Suggestions for Authors
The manuscript molecules-3377818 entitled "Synthesis and Characterization of a Two-station Two-gate Calix[6]arene-based [2]Catenane" by Secchi and co-workers describes the design and synthesis of novel mechanically interlocked molecules, i.e., [2]rotaxane and [2]catenane, based on calix[6]arene derivative. This manuscript is a logical continuation of many years of scientific work by the team of authors. The authors propose and excellently realize approaches to obtain very challenging molecules. The structure of the synthesized compounds was confirmed by spectral methods (NMR spectroscopy and mass-spectrometry). The manuscript is well-written, scientifically significant and contains novelty in the synthesis of [2]catenane based on calix[6]arene. References cited in this manuscript are appropriate and relevant to this study. Overall, the conclusions are valid and reliably proven by physical methods.
I think this paper can be of interest to readers of Molecules. Some minor recommendations to improve the manuscript are given below.
1) I suggest rewriting the abstract according to the journal's recommendations (https://www.mdpi.com/journal/molecules/instructions).
2) Please add the names of the equipment used in the physicochemical studies (NMR spectroscope, mass spectrometer, melting point apparatus, etc.).
3) Mass spectra of all synthesized compounds should be added to the Supplementary Materials.
4) Some abbreviations have no decoding. Please correct this.
5) The quality of Figures 3–5 is poor (the scale and some of the peaks are not clearly visible). Please improve it.
Author Response
Q1. I suggest rewriting the abstract according to the journal's recommendations.
A1. We added a short background section (six text rows) in the first part of the abstract as requested by the journal’s recommendation.
Q2. Please add the names of the equipment used in the physicochemical studies (NMR spectroscope, mass spectrometer, melting point apparatus, etc.).
A2. Done
Q3. Mass spectra of all synthesized compounds should be added to the Supplementary Materials.
A3. Done
Q4. Some abbreviations have no decoding. Please correct this.
A4. Fixed
Q5. The quality of Figures 3–5 is poor (the scale and some of the peaks are not clearly visible). Please improve it.
A5. Fixed
Reviewer 4 Report
Comments and Suggestions for Authors
The manuscript by Secchi and coworkers describes the synthesis of 2-station catenane with 2 stations and 2 gates. The manuscript has an excellent quality and it is well presented. The synthesis of the compounds is very well described with the appropriate characterization made very carefully. The manuscript is very well written with high quality figures, making it very easy to follow. The novelty of the manuscript is good, providing a state-of-the-art catenane with switchable stations in response to stimuli. I recommend the publication of the article in the present form.
Minor comments include the calculated MS mass and the chemical formula. Include the mass of the obtained products in addition to the yield.
Author Response
Q1. … include the calculated MS mass and the chemical formula. Include the mass of the obtained products in addition to the yield.
A1. Fixed